# Insight into Role of Arc Torch Angle on Wire Arc Additive Manufacturing Characteristics of ZL205A Aluminum Alloy

**DOI:** 10.3390/ma17153771

**Published:** 2024-07-31

**Authors:** Bingqiu Wang, Ruihan Li, Xiaohui Zhou, Fuyun Liu, Lianfeng Wei, Lei Tian, Xiaoguo Song, Caiwang Tan

**Affiliations:** 1State Key Laboratory of Precision Welding & Joining of Materials and Structures, Harbin Institute of Technology, Harbin 150001, China; wangbingqiu2001@163.com (B.W.); lrh13722290202@163.com (R.L.); zhouxiaohui1026@sina.com (X.Z.); songxg@hitwh.edu.cn (X.S.); 2School of Materials Engineering, Shandong Institute of Shipbuilding Technology, Weihai 264209, China; 3The Key Nuclear Fuel and Nuclear Materials Laboratory of China, Nuclear Power Institute of China, Chengdu 610213, China; wfenghit@163.com; 4Qingdao Branch of China National Offshore Oil Corporation, Qingdao 266100, China; tianlei@cooec.com.cn

**Keywords:** additive manufacturing, ZL205A aluminum, arc torch angle, microstructure, mechanical property

## Abstract

The arc torch angle greatly affected the deposition characteristics in the wire arc additive manufacturing (WAAM) process, and the relation between the droplet transition behavior and macrostructure morphology was unclear. This work researched the effect of torch angle on the formation accuracy, droplet transition behavior and the mechanical properties in the WAAM process on a ZL205A aluminum alloy. The results suggested that at the obtuse torch angle, part of the energy input was used to heat the existing molten pool, which was optimized for the longer solidification period of the molten pool. Therefore, the greater layer penetration depth at 100° resulted in the improved layer-by-layer combination ability. The obtuse torch angle was associated with the better formation accuracy on the sidewall surface due to the smaller impact on the molten pool, which was influenced by both the arc pressure and droplet impact force. The eliminated pores were optimized for the mechanical properties of depositions at a torch angle of 100°; thus, the tensile strength and elongation attained maximum values of 258.6 MPa and 17.1%, respectively. These aspects made WAAM an attractive mode for manufacturing large structural components on ZL205A aluminum alloy.

## 1. Introduction

As a typical Al-Cu-Mn series casting aluminum alloy, a ZL205A alloy is suitable for producing large structural components with excellent properties, including its light weight, high strength and high toughness [1]. It has been widely applied in military fields such as in rocket bodies and long-range missiles, as well as in aerospace, aviation, etc. [2]. In recent years, with continuous improvements in design indicators for products, the requirements for large aluminum alloy structural components have increased. However, conventional subtractive manufacturing technology causes material waste, and defects in the casting components, including shrunken pores, shrinkage and hot cracks, harm mechanical performance. These aspects led to a reduction in the current qualification rate of the market product; thus, conventional manufacturing technology had to be replaced [3,4,5].

To meet the requirement for high-performance fabrication of large-scale structures, wire arc additive manufacturing (WAAM) technology on a ZL205A aluminum alloy has been developed, and it has shown wide application prospects in military fields [6,7]. Zhou et al. [8] observed that the 205A structural parts fabricated by CMT-WAAM effectively avoided the defects of shrinkage and cracks of casting parts. Ma et al. [9] noted that compared to casting, the ultimate tensile strength of the CMT-WAAM 205A deposited wall showed an enhancement. Liu et al. [10] proved that the yield strength and elongation of 205A parts under CMT-WAAM were 31.3% and 211.8% higher than those of the casting method. However, the plasticity in the vertical direction still limits current applications of ZL205A, with the elongation only increasing to 7.8–16.2%. Furthermore, the formation quality of the deposited parts has been neglected in previous papers on 205A WAAM, which affects the reduction in material waste.

To overcome the above-mentioned difficulties on avoiding subsequent mechanical post-processing and improving the mechanical performance of components, some previous papers have researched aluminum WAAM. The studies mainly focused on process optimization during deposition and the development of a heat source. To achieve even heat accumulation in the directional deposition process, a vertical arc torch was generally utilized in the following WAAM studies. Xue et al. [11] studied the construction of energy gradient for switching weak and strong pulses in double-pulse MIG welding on aluminum. The results showed that controlled arc breakdown was conducive to the welding stability, leading to uniform formation. Cong et al. [12] investigated the fact that compared to conventional CMT, CMT advanced pulse technology was more suitable for manufacturing Al-Cu structures with eliminated porosity due to the proper heat input controllability. Ma et al. [13] proved that due to the enhanced stabilization of both droplet transfer and arc dynamic behavior, a smoother surface of the WAAM depositions was acquired with the addition of a laser. Chen et al. [14] proved that with a reduction in arc length, the impact on the molten pool and the velocity of the droplet moving into the molten pool were greatly decreased. The stability of dynamic molten pool behavior was high during the droplet transition period, resulting in a uniform thickness of the obtained WAAM Al-Si wall. Therefore, it could be inferred that the pulse mode of WAAM and the arc torch with an inclination angle may positively impact the characteristic of the arc, contributing to its enhanced formation accuracy and mechanical performance.

The inclined torch angle was related to various arc characteristics, which had a significant impact on the formation, droplet transfer, arc behavior and heat input on the layers [15,16]. Therefore, as an important thermal control parameter, an inclined arc torch has been proposed to improve the formation accuracy and deposition stability of deposited parts. Su et al. [17] researched the impact of torch angle on the first layer of the deposition during aluminum CMT manufacturing processes. The results revealed that when the torch angle was 90°, the first deposited layer exhibited high porosity due to the larger arc force and smaller buoyancy force. Li et al. [18] researched the impact of torch angle on the molten pool stability of thin wall parts during the steel WAAM process, and the results indicated that depositing with an acute torch angle was capable of enhancing the deposition rate and stability of the molten pool. The aforementioned research has suggested that the arc torch angle is crucial for molten pool behaviors, the formation quality and the mechanical performance of WAAM structures. However, few studies on the relation between droplet transition behavior and macrostructure morphology in WAAM have been reported.

In this study, thin wall parts on a ZL205A aluminum alloy were fabricated by a double-pulse MIG WAAM system with distortion of the torch angle. Dynamics of the molten pool were monitored and recorded by a monitoring system. The impacts of the torch angle on the formation accuracy, droplet transition behavior and mechanical properties were systematically assessed and analyzed. The relation between the droplet transition behavior and the thin wall formation with varying torch angle was explored, which provided an effective mode for fabricating WAAM ZL205A structures.

## 2. Materials and Methods

In this experiment, according to ISO 18273 [19], the ZL205A aluminum alloy provided by Shanghai Dangda Technology Company, Shanghai, China, was accepted as filling wire, and the diameter was 1.2 mm. The wire was deposited on 6061 aluminum alloy plates with a size of 150 mm × 75 mm × 8 mm. Pure argon (99%) was applied as the shielding gas, and the gas flow rate was set as 20 L/min. The chemical composition of the ZL205A filling wire is listed in Table 1.

Figure 1a shows that the double-pulse MIG WAAM system mainly included a KUKA robot system, a Fronius tps 400i CMT welding system and a high-speed camera monitoring system. We explored the deposition methods on ZL205A under MIG and CMT in the pre-experiment stage, while obvious large-size pores and cracks were observed. To reduce the mentioned defects and achieve better printing quality, we then conducted the double-pulse MIG WAAM experiment and selected the optimum deposition parameters. The optimum deposition parameters were chosen and are shown in Table 2. The vertical distance between the torch nozzle center and substrate was kept at 15 mm. The vertical distance from the head of extended filler wire to the substrate was set as 3 mm. The torch angle was defined as the angle between the wire feeding direction and the traveling direction, which ranged from 60° to 120° in this experiment. This was because a discontinuous layer would be observed as the torch angle was less than 60° or greater than 120°. The interlayer waiting time was set as 2 min to control the interlayer temperature. The droplet transition behavior was captured by a high-speed camera (iX Cameras i-SPEED 221 provided by iX Cameras Company, Essex, UK) at different torch angles, and a computer was used for monitoring and analyzing. The high-speed camera axis was tilted at 20° with a horizonal direction. After observing the real-time images, the maximum cross-section arc area was measured using ImageJ 1.53 software. Figure 1b depicts the directional deposition strategy utilized for the deposited wall. The torch angle changed according to the traveling direction. During the double-pulse MIG deposition process, the formation of fish scale could be adopted as a method to evaluate the formation quality.

Prior to the experiment, the oxide film was removed from the substrate through grinding and polishing. Figure 2a suggests the location of the microstructure samples and tensile samples. The test specimens were obtained from the depositions by a wire electrical discharge machine after deposition. The microstructure samples were ground with silicon carbide papers and polished to a mirror finish. Before being observed by the OLYMPUS DSX510 optical microscope (OLYMPUS Company, Tokyo, Japan), a Keller reagent (1 mL HF + 1.5 mL HCl + 2.5 mL HNO_3_ + 95 mL H_2_O) was employed to corrode specimens for 30 s. The samples for sidewall morphology along the height direction were cleaned and observed with a microscope in the three-dimensional mode. Figure 2b presents the dimension of the tensile test specimens vertical to the scanning direction. According to ASTM E8/E8M-24 [20], in order to obtain the effect of the microstructures and defects on tensile properties, the tensile test specimens were examined with an INSTRON room-temperature tensile testing machine provided by INSTRON corporation, Boston, MA, USA, and the loading rate was 1 mm/min. A Zeiss MERLIN compact scanning electron microscope (SEM) provided by Zeiss Company, Oberkochen, Germany, was utilized to obtain morphologies of tensile fracture surface.

## 3. Results and Discussion

### 3.1. Macrostructure Morphology

The macro-morphology had an important impact on the mechanical properties and service performance of the deposited structure. Figure 3 illustrates the surface morphologies and the local magnified images of the depositions obtained by additive manufacturing with a torch angle of 60–120° at every 10°. Severe splashes were observed in the substrate when the torch angle was between 60° and 90°, as shown in Figure 3a–d. Additionally, the adjacent-layer boundary of the sidewall surface was a curve pattern, which was characterized by an uneven fish scale. Especially at the torch angle of 60°, significant molten pool overflow was observed. However, the fish scale of the sidewall surface was shallower and the adjacent-layer boundary exhibited a straight pattern when the torch angle tilted to 100–120°, as presented in Figure 3e–g. This indicated that the molten pool overflow in the additive manufacturing process was obviously suppressed when the torch angle gradually increased to an obtuse angle. Furthermore, although the formation of the thin wall with the same deposited height of 40 mm was effectively changed, the layer height showed little association with the increased torch angle.

To further analyze the impact of the torch angle on the formation quality, Figure 4 presents the three-dimensional height images of the local sidewall surface and the height contours extracted along the building direction. When the torch angle was 60°–90°, a sidewall surface with significant height peaks and height valleys was observed, as shown in Figure 4a–d. However, the fluctuation of the sidewall surface was significantly decreased as the torch angle exceeded 90°, transitioning from a serration pattern to a smooth undulation pattern, as illustrated in Figure 4e–g. This suggests that the increase in the torch angle was beneficial for weakening the height difference between layers on the sidewall surface.

In order to quantitatively assess the surface formation quality, the maximum value of height difference Sz and the surface roughness Sa were extracted from the three-dimensional height graphs, and the statistics are plotted in Figure 5. Compared with the torch angle of 60–80°, both Sa and Sz were evidently decreased when the torch angle exceeded 90°, suggesting that the formation accuracy was significantly improved when the torch angle was an obtuse angle. Especially at the torch angle of 110°, Sa and Sz reached the minimum value of 88 μm and 536 μm, respectively. This indicated that the obtuse torch angle in the deposition process was favorable for reducing material waste during the post-machining process.

To investigate the relation between torch angle and the interlayer combination of additively manufactured structures, Figure 6 shows the cross-section morphologies of the depositions with a torch angle of 60–120°. When the torch angle was less than 90°, the substrate metal was only melted a little, with poor combination of the substrate and the deposited metal, as depicted in Figure 6a–c. However, the melted area on the substrate was enlarged when the torch angle exceeded 90°, as presented in Figure 6d–g. Especially when the torch angle was 100°, the maximum melted area on the substrate was observed, suggesting that the best combination between the substrate and the deposited metal was reached. Additionally, a straight and parallel interface boundary feature was formed at the torch angle of 90°, while obvious offset between the center of the adjacent deposited layer was observed when the torch angle was slightly tilted to 80° and 100°. This was attributed to the deflection of the arc bell shape and the offset between the droplets and the centerline of the previous layer as the torch angle deviated from 90°. However, the interface boundary line transitioned back to the straight pattern when the torch angle continued to decrease to below 80° or increase to above 100°, as illustrated in Figure 6a,b,f,g. This suggested that the offset of the adjacent deposited layer was evidently induced by the slightly inclined torch in the manufacturing process.

Based on the above-mentioned analysis, a better macrostructure of the deposition was achieved with the obtuse torch angle in the thin-walled deposition process due to the higher formation accuracy and the better layer-by-layer combination ability.

### 3.2. Effect of Torch Angle on Droplet Transition Behaviors

The torch angle was associated with the droplet transfer, molten flow behavior and arc shape. Figure 7 presents high-speed pictures of the additive manufacturing process captured with torch angles of 60°, 90°, 100° and 120. The droplet transition process of double-pulse MIG WAAM can be divided into three stages: (i) droplet formation and detachment, (ii) droplet dropping through the air and (iii) droplet entering into the molten pool.

The shortest time for stage (i) of droplet formation and detachment was 1 ms at the torch angle of 90°, while this period increased to 2 ms when the torch angle decreased to 60° and increased to an obtuse angle, as presented in Figure 7a1,a2,b1,c1,c2,d1,d2. This suggested that the rate of droplet growing to detachment was slowed down when the torch angle deviated from 90°. The arc was obviously deformed and compressed with a sharp tail at the bottom of the arc when the torch angle was 60°, resulting in concentrated energy input with the heating point moving forward. However, the arc shape changed to a symmetrical bell shape and a trailing broom shape when the torch angle tilted to 90° and was an obtuse angle, respectively, resulting in a uniform energy input and dispersed energy input on the molten pool, respectively. Moreover, shifts from 90° in the growth direction of the droplet were observed when the torch angle deviated from 90°, suggesting that the forces on the droplet acted along the wire extended direction, as shown in Figure 7a2,c2,d2.

The period for stage (ii) of the droplet dropping through the air was the shortest at the torch angle of 90°, as shown in Figure 7a3,a4,b2,c3,c4,d3–d5. Before sequentially entering the molten pool along the wire extended direction at 60°, the detached droplet was separated into a large-size droplet and a small one, as depicted in Figure 7a3,a4. Then, an intense fluctuation was observed after the large-size droplet entered the molten pool accompanied by the small droplet. As presented in Figure 7b2, when the droplet detached, a thin metal thread between the droplet and the wire was formed at 90°. The droplet then fell into the molten pool vertically with a size similar to that at 60°. As shown in Figure 7c3,c4,d3–d5, the small-size droplet in the droplet pattern detached from the end of the filling wire when the torch angle was an obtuse angle, and the molten pool exhibited a smooth mirror pattern surface without the significant fluctuation. This suggested that the obtuse torch angle was beneficial for the formation and transition of the small-size droplet during the droplet flying period.

Figure 7a5,a6,b3–b6,c5,c6,d6 depicts that the molten pool fluctuation was weakened with an increase in the torch angle in stage (iii) of the droplet entering into the molten pool. A vortex flow pattern was formed and a depression in the molten pool center occurred as the torch angle was 90°. With the droplet entering the molten pool, the metal flowed towards the forefront of the molten pool at 60°, resulting in the most severe molten pool fluctuation. In addition, evident spatters were observed when the torch angle was 90° or less than 90°. However, the molten material moved towards the rear of the molten pool almost without fluctuation when the torch angle increased to an obtuse angle, and the molten pool surface first returned to flatness after the droplet transition cycle ended. This suggested that improved stability of the deposition process was achieved at the obtuse torch angle.

Investigation of the formation mechanism in the additive manufacturing process was of great significance to improve the performance of the deposited structure. The mechanism of the torch angle on the droplet transition process was affected by two key factors: the energy input and the forces. Firstly, we analyzed the energy input on the molten pool. The arc morphology was significantly changed as the torch inclined during the deposition process, as depicted in Figure 7a1,b1,c1,d1, which was related to the energy input. Therefore, Figure 8 summarizes the maximum arc area of the droplet transition cycle at a different torch angle for the following analysis. The arc area was enlarged as the torch angle inclined from 60° to 120°. The molten pool was evenly heated due to the symmetrical arc shape at the torch angle of 90°, contributing to temperature homogenization of the molten pool. The inclination and compression of the arc induced the arc heating point to move forward, resulting in concentrated energy input in front of the molten pool when the torch angle was 60°; thus, a large amount of energy input was used to preheat the deposited metal of the previous layer. As a result, compared with the torch angle of 90°, the energy input in the central region of the molten pool was smaller and the deposited metal was melted at a lower temperature. In addition, due to the characteristics of the double-pulse current, the stage of high energy input and low energy input interacted with the great impact of the arc force, resulting in vibrated and regular molten pool fluctuation [11]. In this case, due to the unstable deposition process, the fish scale on the sidewall surface was easy to form, as presented in Figure 3. The energy input was concentrated at the back of the molten pool when the torch angle inclined to an obtuse angle of 100°. Additionally, the formation of a thin oxide film led to a reduced diffusion rate of heat and mass [21]. Due to the longer length of the molten pool at 100°, the diffusion rate of heat and mass was decreased compared with the 60° and 90° angles, which was beneficial for the persistence time of the molten pool. Furthermore, compared with the 100° angle, the arc action area of 120° on the previous layer was reduced, leading to more heat loss of the molten pool, which was not conducive to metal deposition in the manufacturing process.

In order to explore the characteristics of droplet transition at different torch angles, the forces were analyzed subsequently. Figure 9a illustrates a schematic diagram of the forces during stage (i) of droplet formation and detachment and (ii) of the droplet dropping through the air. During stage (i) of droplet formation and detachment, the main forces acting on the droplet included the gravity force *F_g_*, axial electromagnetic force *F_ema_*, plasma flow force *F_p_* and surface tension force *F_σ_* [22]. The radial electromagnetic force *F_emr_* acted on the droplet necking along the tangential direction. The surface tension force *F_σ_* acting in the opposite wire extended direction prevented droplet detachment [23], while the other forces acting on the droplet accelerated the droplet transition. However, *F_ema_* acted in the opposite wire feeding direction when the arc was compressed [24]. The surface tension force *F_σ_* was calculated using Equation (1):(1)$$F_{\sigma }=2\pi r_{w}\sigma$$
where *r_w_* was the radius of the filling wire and *σ* was the surface tension coefficient (0.28 N/m) [25]. The formula for the gravity force *F_g_* was Equation (2):(2)$$F_{g}=\frac{4}{3}\pi r_{d}^{3}\rho g$$
where *r_d_* was the droplet radius, *ρ* was the droplet density (2.8 g/cm^3^) and *g* was the gravitational acceleration (9.8 m/s^2^). The formula for the plasma flow force *F_p_* was Equation (3):(3)$$F_{p}=\frac{1}{2}C_{p}\pi r_{d}^{2}\rho_{p}KI^{2}$$
where *C_p_* was the coefficient of the plasma flow force (0.45), *ρ_p_* was the arc plasma density [26] and *K* was a scale coefficient [27]. Current *I* could be the arc inner-layer current (*I_core_*) due to the metal vapor column that enveloped the droplet [28]. The axial electromagnetic force F_ema_ could be calculated by Equations (4)–(6):(4)$$F_{ema}=\frac{\mu_{0}I_{core}^{2}}{4\pi }\left(P\left(r\right)+P\left(\theta \right)\right)$$
(5)$$P\left(r\right)=ln\frac{r_{d}}{r_{w}}-\frac{1}{4}$$
(6)$$P\left(\theta \right)=lnsin\theta -\frac{1}{1-cos\theta }+\frac{2}{{\left(1-cos\theta \right)}^{2}}ln\frac{2}{1+cos\theta }$$
where *θ* was the shape coefficient, *μ_0_* was the vacuum magnetic permeability [29], *P(r)* was the morphology coefficient of radius and *P(θ)* was the morphology coefficient of the conduction region angle. With the detachment of the droplet, *θ* was set as 90° because the arc enveloped the bottom surface of the droplet. Thus, the formula could be simplified to Equation (7):(7)$$F_{ema}=\frac{\mu_{0}I_{core}^{2}}{4\pi }\left(ln\frac{r_{d}}{r_{w}}+0.1363\right)$$

The radial electromagnetic force *F_emr_* promoted necking and cutting, which could be expressed as Equation (8):(8)$$F_{emr}=\frac{\mu_{0}I^{2}}{4\pi }$$
where *I* was the current passing through the liquid bridge. Current *I* equated to the peak value of the total current (*I_peak_*) as the liquid bridge broke. With the continuous growth, the resultant force *F* pushed the droplet to detach from the wire along the wire feeding direction. *F* could be represented as Equation (9):(9)$$F{=F}_{p}+F_{ema}{+F}_{g}cos\theta -F_{\sigma }$$
where *F_p_* was the maximum value of plasma flow force under the peak arc current. During stage (ii) of the droplet dropping through the air, the surface tension force *F_σ_* and radial electromagnetic pinch force *F_emr_* disappeared. At this stage, the resultant force *F* acting on the droplet could be calculated by Equation (10):(10)$$F{=F}_{p}+F_{ema}{+F}_{g}cos\theta =ma$$
where *m* was the droplet mass and *a* was the acceleration in this stage. According to the previous works, the magnitude order of the forces acting on the droplet and the droplet necking varied significantly, as listed in Table 3.

Figure 9b–d illustrate the schematic diagram during droplet transition, with various torch angles, in the additive manufacturing process. During stage (i) of droplet formation and detachment, the effect of *F_emr_* on the initial velocity was negligible upon the detachment of the droplet due to its small value. In addition, *F* functioned along the wire feeding direction since *F_g_* was much smaller than the value of *F*, which could be resolved into a horizontal force *F_h_* along the scanning direction and a vertical force *F_v_* along the building direction. When the resultant force *F* reached its maximum with the maximum inner-layer arc current, droplet necking and detachment were performed. When the torch angle was 90° or less than 90°, the values of *F_p_*, *F_g_* and *F_ema_* acting on the droplet were all larger than those of greater than 90° with the same *F_σ_*. However, the direction of *F_ema_* was opposite at the acute angle; thus, the value of *F* was seldom changed compared with the obtuse angle. Eventually, the value of *F* reached the maximum at 90°. Compared with the acute and obtuse angles, the droplet detachment rate with the vertical torch angle was accelerated, and the initial velocity was increased with a larger droplet size when the droplet was about to fall off.

The *F_g_* was much smaller than the resultant force *F* in stage (ii) of the droplet dropping through the air; thus, the droplet dropped along the wire feeding direction. When the torch angle was 90°, there was no horizontal component of the force *F_2_* in the droplet flight process; thus, the droplet position flying into the molten pool was right below the arc torch, as presented in Figure 9c. In addition, the flight distance was the shortest, which was beneficial for the shorter transition time in stage (ii), which can be confirmed in Figure 7b2. As depicted in Figure 9b, the droplet pointed towards the forefront of the molten pool and first contacted the previous layer metal when the torch angle was 60°. The deposited material in the molten pool flowed forward and tended to accumulate in the front of the molten pool along the scanning direction. As a result, significant deposition resistance of the molten pool to move forward was caused by the metal of the previous layer, which efficiently prevented the deposition process. Furthermore, the molten metal in the molten pool was pushed away from the previous layer, which was induced by the pushing force of the arc pressure, leading to a decreased penetration depth compared with the 90° angle, as depicted in Figure 6a–c. Due to the increased flight distance and deceased flight velocity during this stage, the flight time was increased compared with the 90° angle, which can be confirmed in Figure 7a3,a4. Additionally, the temperature in the forefront of the molten pool was higher due to the uneven energy input compared to the tail; thus, the gas emission from the molten pool was increased. Accompanied by severe droplet scattering owing to the repellent arc force and repellent electro-magnetic force, severe spatters during the transition process occurred [31].

As presented in Figure 9d, when the torch angle exceeded 90°, the horizontal force *F_h_* in the opposite scanning direction acted on the droplet. Thus, the droplet fell into the rear part of the molten pool, dragging the metal to flow towards the back of molten pool. The impinging impact on the molten pool was influenced by both the droplet impact force and arc pressure [17], which was reduced when the torch angle exceeded 90°. The arc pressure *F_arc_* could be expressed as Equation (11) [32]:(11)$$F_{arc}=\frac{uI^{2}}{8\pi }(1+2\ln\left(\frac{R_{1}}{R_{2}}\right))$$
where *u* was the vacuum magnetic permeability, *R_1_* was the arc radius at the filler wire and *R_2_* was arc radius contacting with the molten metal. The droplet gravity and arc force effected the droplet impact [33]. Therefore, molten fluctuation during this stage was reduced. The impact on the molten pool and resistance became smaller during the deposition process; thus, the sidewall surface of the deposited wall tended to be smoother and more regular with a shallow fish scale [34], which can be confirmed in Figure 3e–g. When the torch angle was 100°, part of the energy input was utilized to heat the existing molten pool, contributing to the enlarged contact area of the adjacent molten pool and longer presence compared to the 90° angle. This resulted in a greater layer penetration depth at 100° in the backward position of the solidified region, as suggested in Figure 6e. When the torch angle increased to 120°, the ability of the arc to heat backward was increased and the impact force was decreased, resulting in a reduction in penetration depth [33], as suggested in Figure 6g.

### 3.3. Microstructure and Mechanical Properties

Figure 10 illustrates the stress–strain curves as well as the tensile strength and elongation under different torch angles of 60°, 90°, 100° and 120° in the vertical tensile test. As the torch angle increased to 100°, the tensile strength and elongation were higher than those of 60° and 90°, reaching 258.6 MPa and 17.1%, respectively. However, when the torch angle continued to increase to 120°, the tensile strength and elongation were significantly reduced. This indicated that a slightly increased obtuse angle of 100° could improve the mechanical properties of the depositions.

The microstructure of the deposited structure was greatly changed under the function of arc plasma at different torch angles. To further investigate the relation between the heat input and the microstructure features, grain morphology images with torch angles of 60°, 90°, 100° and 120° were collected and the grain size statistics were plotted, as illustrated in Figure 11. The grain size increased when the torch angle changed from 60° or 90° to 100°, while it decreased at 120°. Due to the increased arc action area, the molten pool solidification time was relatively prolonged. This was beneficial for the formation of large equiaxed grain. Eventually, the grain size at the torch angle of 100° was larger compared to that at 90°. However, part of the energy input was used to preheat the forward position and the energy input applied in the weld pool decreased at the torch angle of 60°; thus, the grain size was smaller compared with that of 90°. As the torch angle reached 120°, the energy input was too dispersed to prolong the solidification time. Therefore, equiaxed grains with a large grain size were formed when the torch angle was 100°.

Figure 12 shows the statistical chart of the number of large-size pores and presents the morphologies of the structures obtained under a torch angle of 60°, 90°, 100° and 120°. The large-size pore number was notably high at torch angles of 60° and 90°, of 11 and 9, respectively, and there are no pores at 100°. However, the number of pores rose once more when the torch angle further increased to 120°. Large-size pores distributed in the longitudinal sections were observed especially at the torch angle of 60°, mainly concentrated in the interlayer region. The presence of pores decreased the load carrying capacity, which raised the possibility of fracture. The number of pores was effectively suppressed at the torch angle of 100°, which was beneficial for better performance. This was because the backward heating point of the molten pool of 100° slowed the solidification time, which provided more time for pores to escape. The inclination of the arc torch resulted in a change in the energy input; thus, the grain size and the porosity were affected by the solidification time. Therefore, although the grain was coarse, the porosity was effectively suppressed at the torch angle of 100°, contributing to the higher mechanical properties.

To further analyze the mechanism of the torch angle on mechanical properties, Figure 13 shows the microstructure of the tensile fracture surface after the tensile test in the vertical direction and local magnified images with different torch angles. As presented in Figure 13a, large-size dimples were observed with polymerization features of dense micropores when the torch angle was 60° and exhibited typical ductile fracture characteristics. As presented in Figure 13b, the number of micropores distributed in the fracture surface was decreased when the torch angle increased to 90° with ductile fractures. Moreover, the dimple depth at the torch angle of 90° became shallower compared with the 60° angle. When the torch angle increased to an obtuse angle of 100°, the micropores disappeared and the dimples became more dense, smaller and deeper, characterized by ductile fracture characteristics, as presented in Figure 13c. When the torch angle increased to 120°, in addition to dimple aggregation with micropores, cleavage was also observed, suggesting that the fracture surface exhibited a ductile–brittle composite fracture, as depicted in Figure 13d. This indicated that the dimples became smaller and deeper when the torch angle increased to an obtuse angle and reduced micropores were beneficial for mechanical properties with a slightly inclined obtuse torch angle of 100°.

Under the inclined arc torch, the WAAM technology was characterized by the forces on the droplet and different thermal distributions on the molten pool. Therefore, the formation quality was improved with the reduction in the molten fluctuation and scanning resistance force, and the mechanical properties were increased with the declined porosity. To obtain better thin-walled structures, additively manufactured depositions with a torch angle of 100° were achieved with a higher formation accuracy, more stable deposition process and higher mechanical properties.

## 4. Conclusions

In this study, ZL205A deposited walls with different torch angles of 60°–120° were developed under WAAM. We conducted a systematic examination of the effect of torch angle on the formation accuracy, droplet transition and mechanical properties of the WAAM thin walls. From this work, conclusions can be drawn as follows:As the torch angle increased to 100–120°, the layer-by-layer combination ability was improved in the thin-walled deposition process. The offset of the adjacent deposited layer in the cross-sections was evidently induced by slightly inclined torch angles of 80° and 100°. The formation accuracy on the sidewall surface was increased when the torch angle increased to 100–120°, with the reduced surface roughness Sa varying from 88 μm to 121 μm.At the acute and obtuse torch angles, part of the energy input was used to heat the deposited metal and the existing molten pool, respectively, contributing to the longer presence of the molten pool at the obtuse torch angle. When the torch angle increased from 60° and 90° to 100°, the grain size was increased due to the increased solidification time and decreased again at the torch angle of 120°.The obtuse torch angle was beneficial for small-size droplet formation and high frequency transition, contributing to improved deposition stability. The droplet impact force and arc pressure both influenced the impact on the molten pool, leading to severe molten fluctuation at the acute torch angle. At the acute torch angle, the molten pool metal accumulated in the front and induced the molten pool resistance to move forward, which prevented the deposition process.Although the grain was coarse, the reduced pores were optimized for mechanical properties of the depositions with a slightly inclined obtuse torch angle of 100°. At the torch angle of 100°, the elongation and tensile strength reached 17.1% and 258.6 MPa of the maximum values, respectively.

## Figures and Tables

**Figure 1 materials-17-03771-f001:**
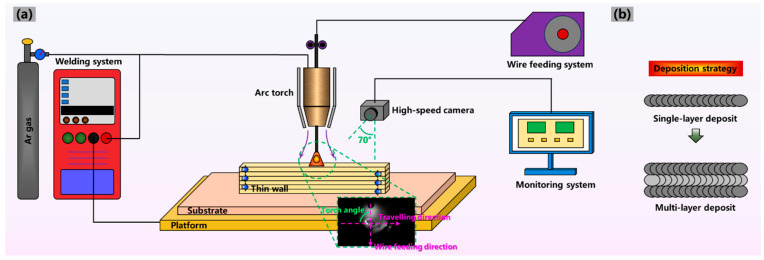
Schematic diagram of (**a**) the double-pulse MIG WAAM system and (**b**) deposition strategy.

**Figure 2 materials-17-03771-f002:**
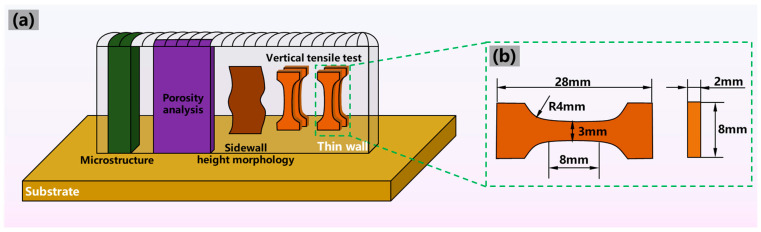
(**a**) The positions of test specimens and (**b**) the dimension of tensile test specimens in the vertical direction.

**Figure 3 materials-17-03771-f003:**
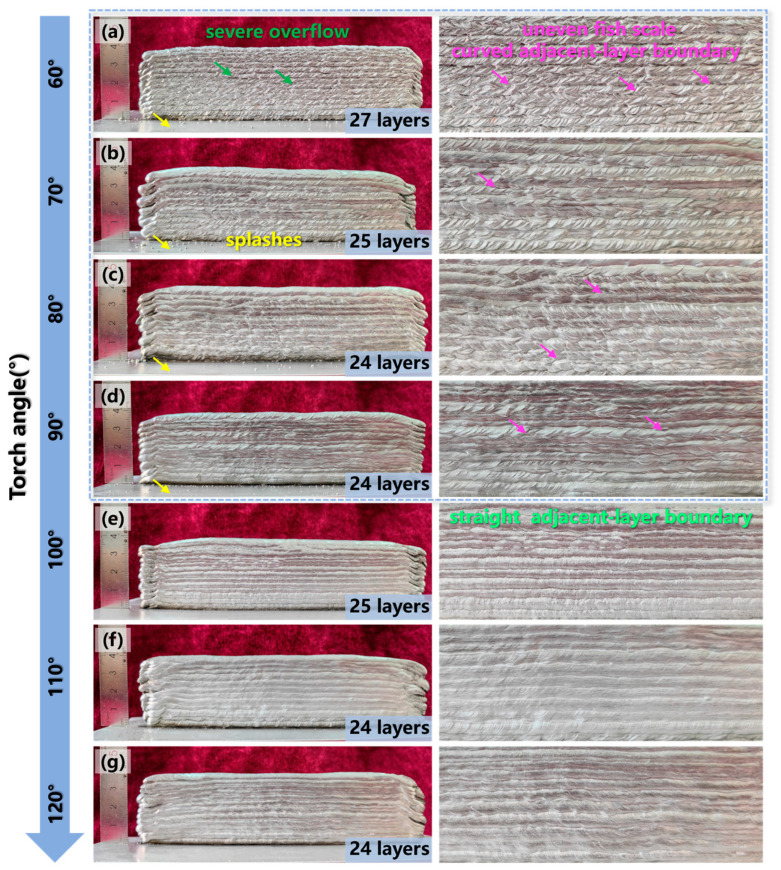
Surface morphologies and details of the depositions with a torch angle of (**a**) 60°, (**b**) 70°, (**c**) 80°, (**d**) 90°, (**e**) 100°, (**f**) 110°, (**g**) 120°.

**Figure 4 materials-17-03771-f004:**
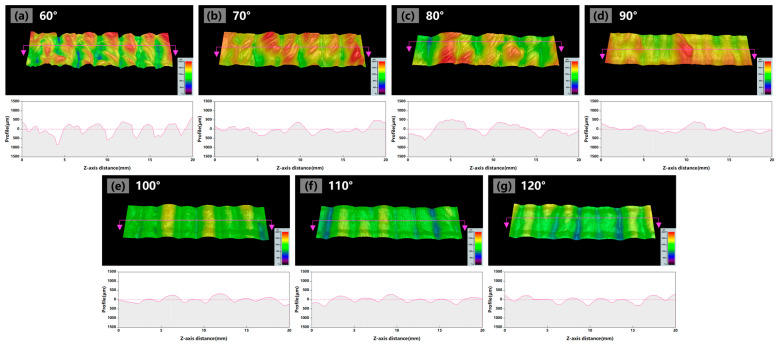
Sidewall surface three-dimensional morphologies and height contours with a torch angle of (**a**) 60°, (**b**) 70°, (**c**) 80°, (**d**) 90°, (**e**) 100°, (**f**) 110°, (**g**) 120°. The red lines with arrows pointed the extracted direction of the height contours.

**Figure 5 materials-17-03771-f005:**
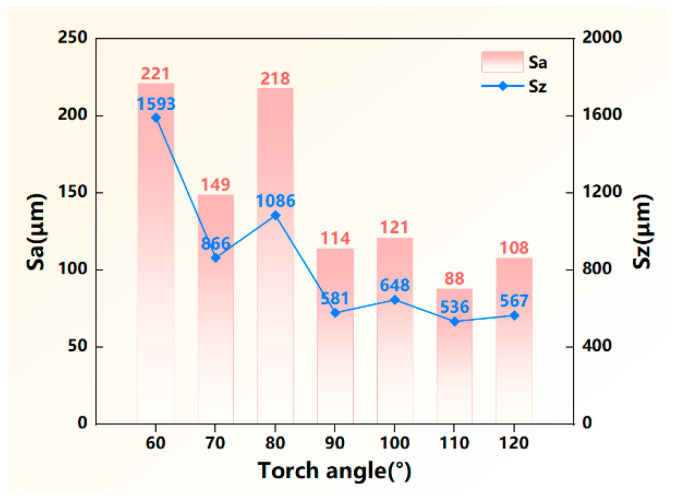
Characteristics of the sidewall formation quality of the deposited wall.

**Figure 6 materials-17-03771-f006:**
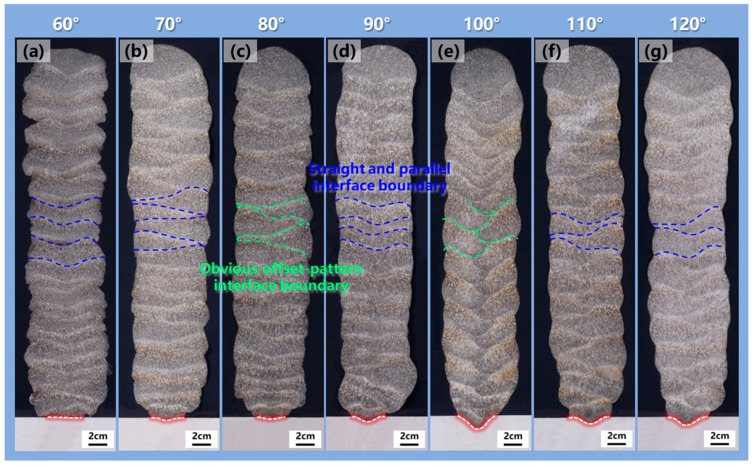
The cross-section morphologies of the depositions with a torch angle of (**a**) 60°, (**b**) 70°, (**c**) 80°, (**d**) 90°, (**e**) 100°, (**f**) 110°, (**g**) 120°. The red dashed line was the fusion line between the deposited metal and the substrate metal.

**Figure 7 materials-17-03771-f007:**
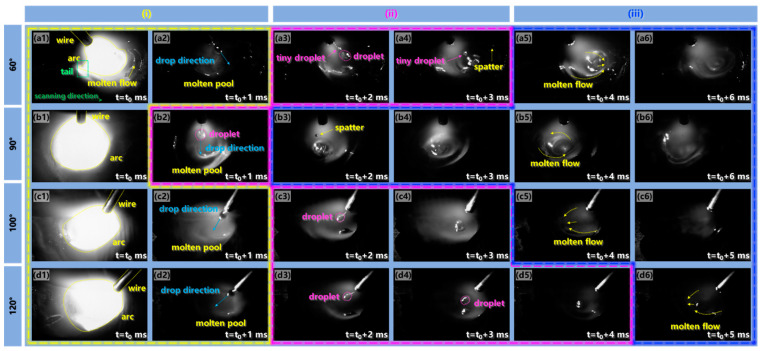
High-speed video frames of the WAAM process with a torch angle of (**a1**–**a6**) 60°, (**b1**–**b6**) 90°, (**c1**–**c6**) 100°, (**d1**–**d6**) 120°.

**Figure 8 materials-17-03771-f008:**
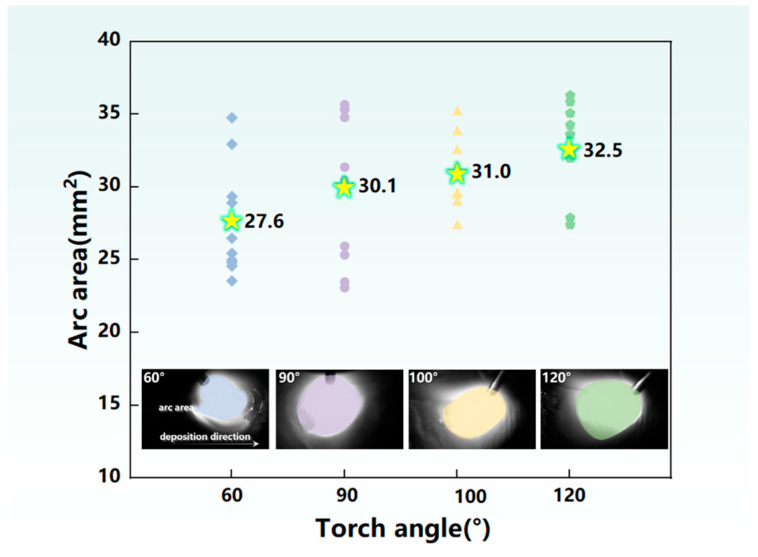
Statistics of the maximum arc area. The value of star symbol represents the average value of scatter data at each torch angle.

**Figure 9 materials-17-03771-f009:**
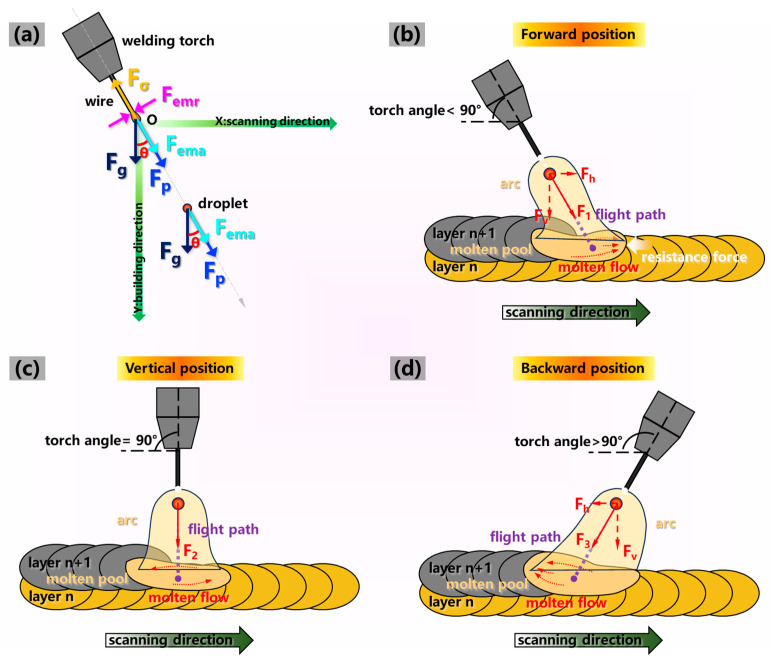
(**a**) The forces during droplet transition and the WAAM process with a (**b**) forward position, (**c**) vertical position and (**d**) backward position.

**Figure 10 materials-17-03771-f010:**
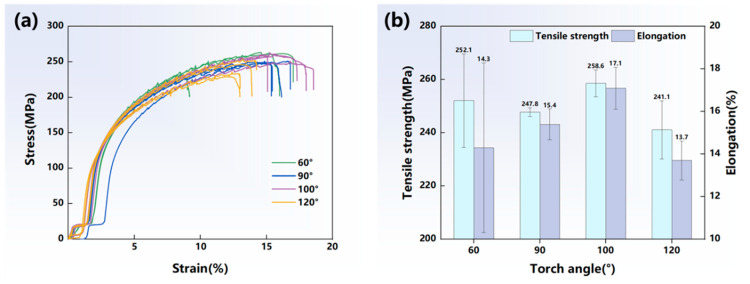
(**a**) Stress–strain curves; (**b**) vertical tensile strength and elongation.

**Figure 11 materials-17-03771-f011:**
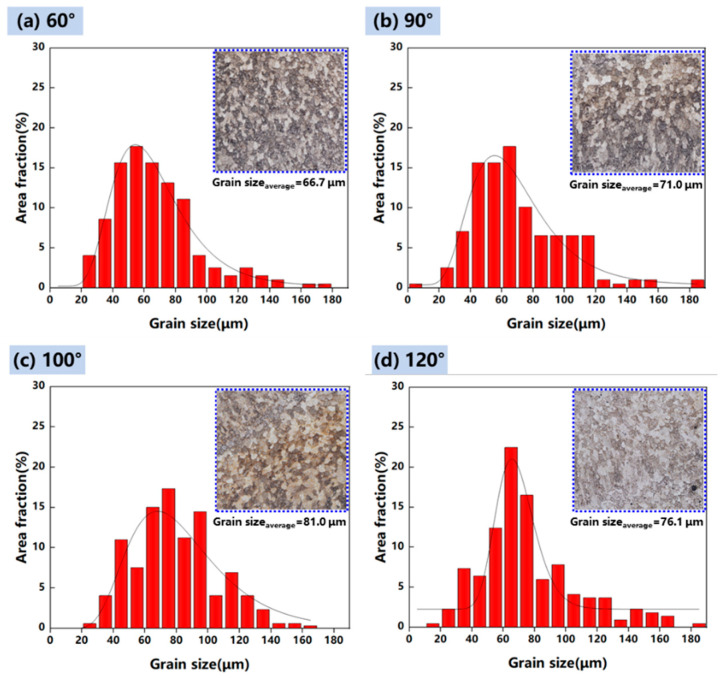
Grain size statistics of samples and images of grain distribution with a torch angle of (**a**) 60°, (**b**) 90°, (**c**) 100°, (**d**) 120°.

**Figure 12 materials-17-03771-f012:**
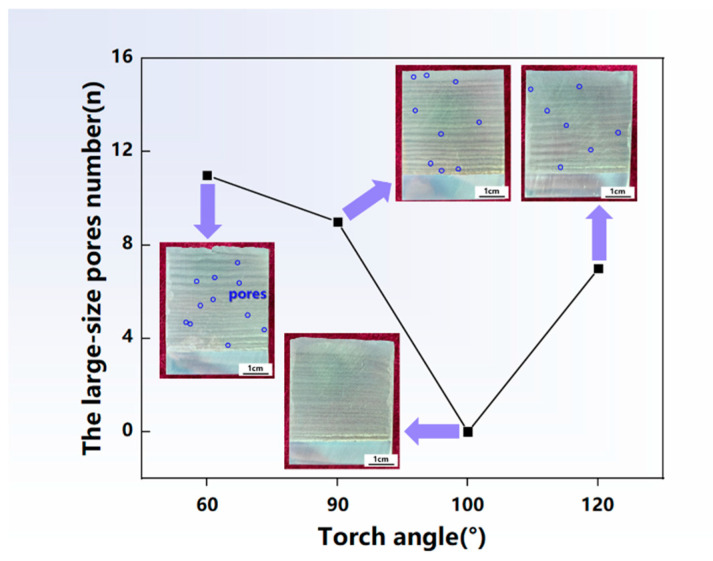
The statistical chart of the large-size pore number and the longitudinal section morphologies of the depositions. The purple arrow points to the section morphologies associated with the data of pores number.

**Figure 13 materials-17-03771-f013:**
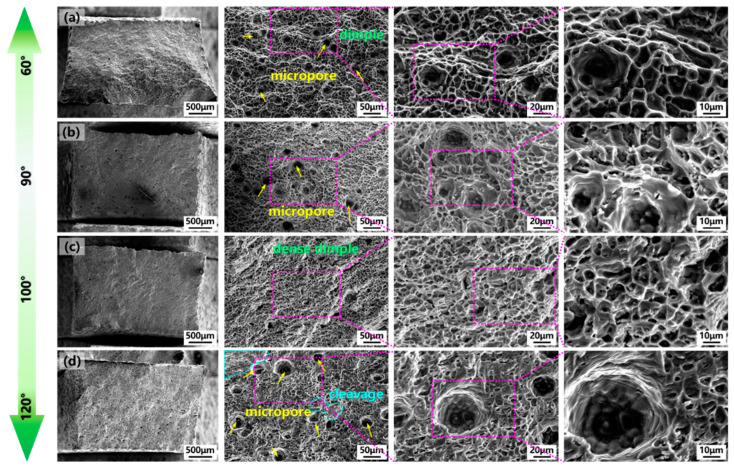
Tensile fracture surface images and magnified view of vertical tensile samples with torch angle of (**a**) 60°, (**b**) 90°, (**c**) 100°, (**d**) 120°.

**Table 1 materials-17-03771-t001:** Chemical compositions of ZL205A (wt.%).

Material	Cu	Mn	Ti	Cd	Zr	B	V	Al.
ZL205A	4.99	0.40	0.14	0.19	0.17	0.03	0.14	Bal.

**Table 2 materials-17-03771-t002:** Experimental parameters of double-pulse MIG WAAM.

Travelling Speed (m/min)	Deposition Current (A)	Wire Feeding Speed (m/min)	Wire Feeding Speed Amplitude (m/min)	Double-Pulse Frequency (Hz)	Pulse Correction	Arc Length Correction
0.4	80	4.8	1.0	10	−10.0	10

**Table 3 materials-17-03771-t003:** Order of magnitude estimations of the forces.

Forces (N)	*F_σ_*	*F_g_*	*F_p_*	*F_ema_*	*F_emr_*
Order of magnitude	10^−3^	10^−4^–10^−3^	10^−4^–10^−3^	10^−3^–10^−2^	10^−4^–10^−3^
Reference	Li et al. [28]	Li et al. [27]	Li et al. [27]	Zhao et al. [30]	Zhao et al. [30]

## Data Availability

The raw data supporting the conclusions of this article will be made available by the authors on request.

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
