# Peer review of "Insight into Role of Arc Torch Angle on Wire Arc Additive Manufacturing Characteristics of ZL205A Aluminum Alloy"

_materials, 2024, doi:10.3390/ma17153771_

Round 1
Reviewer 1 Report
Comments and Suggestions for Authors
The authors investigated the effect of torch angle on the processing, microstructure and mechanical properties of ZL205A aluminum alloy prepared with WAAM. It's rather interesting and suitable for publication with materials.
Below are my comments.
1. The torch angle in the schematic drawing of Figure 1 is very unclear. That's the most import concept of this manuscript, requiring emphasis.
2. Figure 2, it makes the schematic easier to understand if a series of overlaid circles were added to the top of the cuboid. It feels so out of context without them.
3. Page 9~10, the authors presented detailed mathematical expressions of all the forces but no quantitative discussion was done. If keeping all those equations, how about adding the order of magnitude estimations by using the data in the current work?
4. Figure 9, the color selections of some arrows are really poor, which makes it very vague.
5. For pores at different torch angle, are there quantitative measurements?
Reviewer 2 Report
Comments and Suggestions for Authors
The paper contains information adequate to justify publication. The English language is generally of good quality, but at some places corrections are necessary. A regular spelling and grammar check should be adequate. The referenced literature is relevant, up to date, and the number of references is sufficient. The paper's arguments are built on a suitable base of experimentation and knowledge. The results are presented clearly. The discussion is professionally written, with emphasis on the proper analysis and understanding of the experimental results.
Comments below are given with respect to the location in the PDF manuscript associated with line number:
1. Line 13 (page 1, Abstract): ‘… effectively affected …’ (marked in yellow)
Comment: Use of 'effectively' and 'affected' together is inconvenient. Perhaps: 'had effects on ...'
2. Line 108 (page 3): ‘… titled ...’ (marked in yellow)
Comment: Perhaps: 'tilted'
3. Lines 232 (page 8): ‘… was …’ (marked in yellow)
Comment: The word ‘was’ is unnecessary.
4. Lines 246 (page 8): ‘… effectively affected …’ (marked in yellow)
Comment: Please see my first comment ... or ... 'Two key factors had effected the mechanism of ... transition process: the energy input and the forces.'
5. Equation 4 (page 10): ‘P(r) + P(q)’ (marked in yellow)
Comment: P(r) and P(q) have not been defined or described in the text.

Reviewer 3 Report
Comments and Suggestions for Authors
This study presents the impact of torch angle on the formation accuracy, droplet transition behavior and the mechanical properties in the wire arc additive manufacturing (WAAM) process of ZL205A aluminum alloy. The article is well organized and the quality of the images used is decent. But the high similarity percentage to previous articles, not presenting the stress-strain results is the main challenge of the article. In addition, all of the following should be considered.
The abstract can be presented more attractively and emphasize the novelty of the article, especially the printing mechanism.
Although the introduction is very long, it is written very superficially. Most of the mentioned paragraphs are general and this section should be corrected.
Use the following papers. Review of selective laser melting of magnesium alloys: Advantages, microstructure and mechanical characterizations, defects, challenges, and applications. Laser powder bed fusion of Alumina/Fe–Ni ceramic matrix particulate composites impregnated with a polymeric resin. The high temperature flow behavior of additively manufactured Inconel 625 superalloy.
The process of selecting printing parameter rane should be explained. Explain how to choose optimum printing parameters. How is the printing quality checked? Explain more about the printing process.
Provide the ASTM standard and the conditions of the tensile test. Add error bar to quantitive results.
How is the reproducibility of the results as well as the quality of the printing parts determined?
The results of mechanical properties should be presented in the form of stress-strain diagram.
The elongation report above 35% for the printed part is not consistent with the quality provided. How are these results justified?
The main challenge of the article is the very high percentage of similarity to the literature (25%), which has caused all parts of the article to be copies of previous sources. For this purpose, the article should be fundamentally edited.
What is the name of the used aluminum alloy? Mention the trade name and its standard.
Comments on the Quality of English Language**
Round 2
Reviewer 3 Report
Comments and Suggestions for Authors
Accept in present form.